# Benzimidazol-2-ylidene Silver Complexes: Synthesis, Characterization, Antimicrobial and Antibiofilm Activities, Molecular Docking and Theoretical Investigations

Uğur Tutar [1], Cem Çelik [2], Elvan Üstün [3], Namık Özdemir [4], Neslihan Şahin [5,*], David Sémeril [6,*], Nevin Gürbüz [7] and İsmail Özdemir [7]

1. Department of Botanica, Faculty of Pharmacy, Cumhuriyet University, Sivas 58140, Turkey
2. Department of Medical Microbiology, Faculty of Medicine, Cumhuriyet University, Sivas 58140, Turkey
3. Department of Chemistry, Faculty of Art and Science, Ordu University, Ordu 52200, Turkey
4. Department of Mathematics and Science Education, Faculty of Education, Ondokuz Mayıs University, Samsun 55139, Turkey
5. Department of Science Education, Faculty of Education, Cumhuriyet University, Sivas 58040, Turkey
6. Synthèse Organométallique et Catalyse, UMR-CNRS 7177, Strasbourg University, 67008 Strasbourg, France
7. Drug Application and Research Center, İnönü University, Malatya 44280, Turkey
* Correspondence: neslihan@cumhuriyet.edu.tr (N.Ş.); dsemeril@unistra.fr (D.S.); Tel.: +903464874656 (N.Ş.); Tel.: +33-(0)3-6885-1550 (D.S.)

**Abstract:** Five silver(I) complexes, namely chloro[1-methallyl-3-benzyl)benzimidazol-2-ylidene]silver (**6**), chloro[1-methallyl-3-(2,3,5,6-tetramethylbenzyl)benzimidazol-2-ylidene]silver (**7**), chloro[1-methallyl-3-(3,4,5-trimethoxylbenzyl)benzimidazol-2-ylidene]silver (**8**), chloro[1-methallyl-3-(naphthylmethyl)benzimidazol-2-ylidene]silver (**9**), and chloro [1-methallyl-3-(anthracen-9-yl-methyl)benzimidazol-2-ylidene]silver (**10**), were prepared starting from their corresponding benzimidazolium salts and silver oxide in 71–81% yields. A single-crystal X-ray structure of **7** was determined. These five Ag-NHC complexes were evaluated for their antimicrobial and biofilm formation inhibition properties. Complex **10** exhibited high antimicrobial activities comparable to those obtained with standard drugs such as Fluconazole in contact with *Staphylococcus aureus*, *Enterococcus faecalis*, *Escherichia coli*, *Acinetobacter baumannii,* and *Candida albicans*. The latter complex has been shown to be very efficient in antibiofilm activity, with 92.9% biofilm inhibition at 1.9 µg/mL on *Escherichia coli*. Additionally, the molecules were optimized with DFT-based computational methods for obtaining insight into the structure/reactivity relations through the relative energies of the frontier orbitals. The optimized molecules were also analyzed by molecular docking method against DNA gyrase of *Escherichia coli* and CYP51 from *Candida albicans*.

**Keywords:** silver; benzimidazol-2-ylidene; antimicrobial activity; antibiofilm activity; molecular docking

## 1. Introduction

*N*-heterocyclic carbenes (NHCs), one of the most important families of organic ligands for coordination chemistry, are cyclic compounds composed of a divalent carbon atom that bonded at least to one nitrogen atom. Popularized by the groups of Bertrand and Arduengo in the early 1990s [1,2], NHCs have prominent chemical properties such as strong binding ability and easy modulation of electronic and steric properties. In addition, the easy grafting of substituents on the NHC allows fine-tuning of their physical properties, such as solubility or steric hindrance, and even the introduction of chiral groups. After coordination with transition metals, the carbene–metal bond is generally strong and exhibits high stability in various environments [3,4], which has allowed important applications in many fields in catalysis [5–8] and biochemistry, such as anticancer, antimicrobial, antioxidant, and antimalarial [9–13].

Antimicrobial research is another important area where NHC–silver complexes have been particularly successful [14,15]. Before the discovery of modern antibiotics, silver was the most important tool for fighting bacterial infection [16,17]. Due to the increasing growth of highly resistant pathogenic microorganisms, in particular for bacterial biofilms [18], which are often pathogenic and can cause, as revealed by the National Institutes of Health, 65% of microbial and 80% of chronic infections, there is an urgent need for new antibiotics [19,20]. However, in recent years, the approval of new antimicrobial agents has declined, motivating the use of new silver-based complexes or nanoparticles as antibacterial and antibiofilm agents [21–26], even if the presence of aromatic substituents on the ligand allows the lipophilic NHC-silver complexes to inhibit more effectively the formation of biofilm [27]. As examples, Şahin and co-workers demonstrated that bis-alkylated-NHC-silver complexes could display important antibiofilm activities. As examples, bromo[1-methallyl-3-(4-ter-butylbenzyl)-5,6-dimethylbenz-imi- dazole-2-ylidene]silver (**A**) [28] and bromo[1-allyl-3-(4-tert-butylbenzyl)-5,6-dimethyl- benzimidazole-2-ylidene]silver (**B**) [29] could reduce biofilm formation by up to 90% at a concentration as low as 1.9 μM for a large variety of fungi, Gram-positive, and Gram-negative bacteria (Figure 1).

**Figure 1.** Example of silver complexes **A** and **B** exhibiting antibiofilm activities.

Density Functional Theory (DFT) is a useful computational method and provides reasonable predictions of various properties of molecules [30]. Reactivity descriptors are quantitative measures that give insights into how a chemical system behaves in chemical reactions through the relative energies of the frontier orbitals. These descriptors help to understand the reactivity of molecules, which is crucial in drug discovery [31,32]. Molecular docking is also a computational technique used to study the interactions between a ligand and a macromolecule such as an enzyme or protein. One of the most important applications of molecular docking is drug discovery [33,34]. The method helps recognize the potential drug candidates, which can bind to a target protein to influence its activity.

In previous work, optimized Ag–NHC complexes were analyzed by molecular docking methods against Quorum-Quenching N-Acyl Homoserine Lactone Lactonase, SarA, and Malate Synthase A, and useful information was obtained regarding the interaction types and strength of the molecules [12] In continuation of our quest to develop new biologically active silver–NHC complexes, we reported the synthesis of five unseen complexes as breast anticancer, antimicrobial and antibiofilm activities. The relative energies of the frontier orbitals of the optimized molecules were used to gain insight into the reactivity by the chemical reactivity descriptor. The molecules were also analyzed using the molecular docking method against DNA gyrase of *Escherichia coli* and CYP51 from *Candida albicans*.

## 2. Results and Discussion

### 2.1. Synthesis and Characterization of Ag(I)-NHC Complexes

The synthesis of the five silver complexes [35], firstly, required the preparation of the benzimidazolium salts **1–5**, respectively, in which the benzimidazole ring is substituted with a methylallyl and five different arylmethyl moieties, namely benzyl (**1**), 2,3,5,6-tetramethylbenzyl (**2**), 3,4,5-trimethoxybenzyl (**3**), naphthylmethyl (**4**), and anthracen-9-

yl-methyl (**5**). These salts were synthesized according to the reported procedure [36–38] in two steps: first, the nucleophilic of methylallyl chloride on the benzimidazole scaffold, followed by the alkylation of the latter with arylmethyl reagents. The silver complexes (**6–10**) were finally obtained by a reaction of Ag$_2$O with the benzimidazolium salts (**1–5**) in dichloromethane for 24 h at room temperature, protected from light with an aluminum foil (Scheme 1).

**Scheme 1.** Synthesis of silver complexes **6–10**.

The Ag(I)–NHC complexes were isolated in 71–81% yields and were stable to air and moisture in solid and solution but sensitive to light. Therefore, dark conditions were required both for their synthesis and for their applications. The structures of the Ag(I)–NHC complexes were established by FT-IR, multinucleus NMR spectroscopy ($^1$H and $^{13}$C), and LC-MS spectroscopy (see the experimental section and Supplementary Materials).

The $^1$H NMR spectra revealed the disappearance of the acidic proton (NCHN, sharp singlet) from the imidazolium salts at around 11 ppm, except for **5,** for which the signal shifted at 9.21 ppm due to the presence of the electron donor anthracene moiety. In the $^{13}$C NMR spectra of imidazolium salts **1–5**, the NCHN signals appeared in the range 142.1–144.5 ppm while in the corresponding Ag(I)–NHC complexes, the NCN carbon peaks were observed at 189.3, 187.8, and 189.0 ppm for **6**, **7** and **8**, respectively (Figure 2) as broad singlets and not as expected doublet of doublets due to coupling with the two NMR active silver isotopes ($^{107}$Ag and $^{109}$Ag). According to Lin and co-workers, the lack of coupling could be due to the fluxional behavior of the complexes on the NMR time scale [39]. Note that, as in a significant number of silver(I) complexes, these signals could not be observed for complexes **9** and **10** [40] (Table 1).

LC-MS spectra of the silver complexes **6–10** displayed molecular ion peaks corresponding to the [Ag(NHC)$_2$]$^+$ cations at *m/z* = 631.3, 745.4, 813.3, 731.3 and 833.4, respectively. The observation of these bis-carbenic complexes demonstrates the lability (lower Ag-C bond) of the carbene moieties, which is responsible for the reported equilibrium between the [AgX(NHC)] and [Ag(NHC)$_2$]$^+$ [AgX$_2$]$^-$ complexes [41,42]. The formation of [Ag(NHC)$_2$]$^+$ [AgX$_2$]$^-$ complexes took place under the sampling conditions used for LC-MS [43]. In solution, the formation of the latter complexes was not observed.

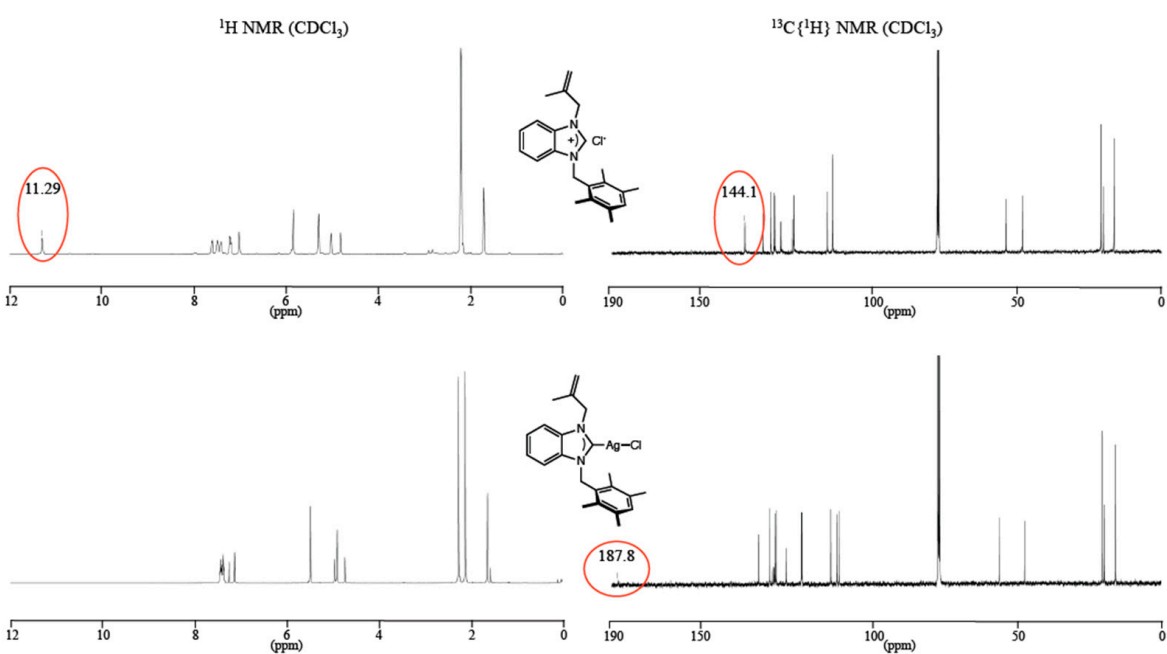

**Figure 2.** NMR spectra ($^1$H and $^{13}$C) of benzimidazolium salt **2** (**top**) and its corresponding silver complex **7** (**bottom**).

**Table 1.** Spectroscopic data of benzimidazolium salts **1–5** [36–38] and their corresponding silver complexes **6–10**.

| | Benzimidazolium Salt | | | | Silver Complex | |
|---|---|---|---|---|---|---|
| | FT-IR $\nu$(CN) (cm$^{-1}$) $^1$ | $^1$H NMR NC*H*N (ppm) | $^{13}$C NMR NC*H*N (ppm) | | FT-IR $\nu$(CN) (cm$^{-1}$) $^1$ | $^{13}$C NMR N*C*(Ag)N (ppm) |
| **1** | 1557 | 11.86 | 144.2 | **6** | 1397 | 189.3 |
| **2** | 1557 | 11.29 | 144.1 | **7** | 1396 | 187.8 |
| **3** | 1558 | 11.74 | 144.1 | **8** | 1392 | 189.0 |
| **4** | 1556 | 11.94 | 144.5 | **9** | 1392 | Not observed |
| **5** | 1561 | 9.21 | 142.2 | **10** | 1394 | Not observed |

$^1$ CN stretching frequency, maximal peak intensity.

The benzimidazolium salts **1–5** revealed in their FT-IR spectra a specific $\nu$(C=N) band between 1557 and 1560 cm$^{-1}$. After silver complexes formation, no peaks were detected in this region, and a downshifting of about 165 cm$^{-1}$ for the $\nu$(C=N) bands was observed (1392–1397 cm$^{-1}$). The decrease in the N=C stretching frequency was due to the electropositive silver atom that attracts the electron density and causes the weakening of the C=N bond [44]. The shift of the $\nu$(C=N) bands could be considered as an indicator of silver complex formation.

### 2.2. Single-Crystal X-ray Diffraction Studies

Coordination of the *N*-heterocyclic carbene to the silver atom and formation of the neutral (NHC)Ag-Cl type monomeric complexes was confirmed by a single-crystal X-ray diffraction study realized on complex **7**. The complex crystallizes in the monoclinic space group C2/c (Figure 3).

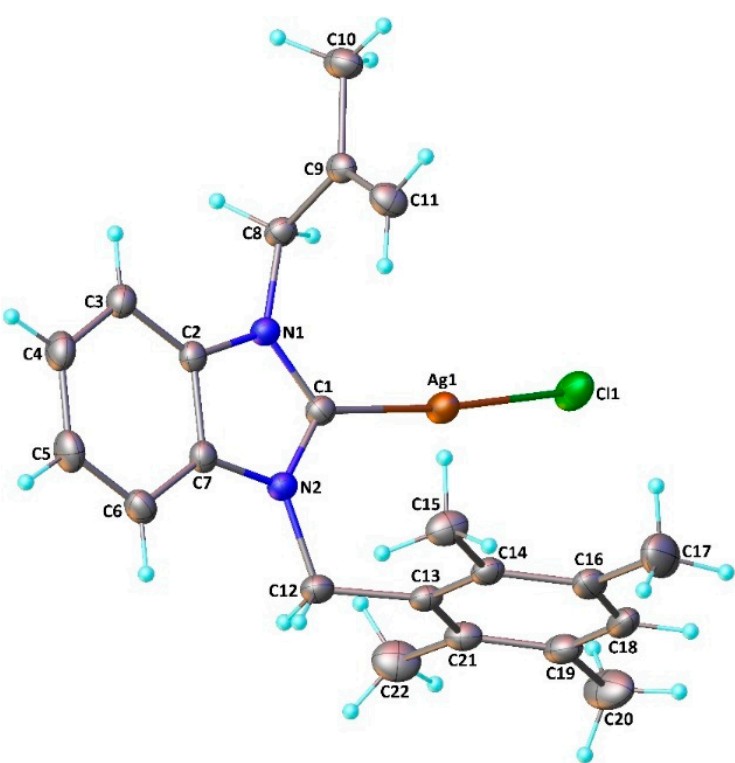

**Figure 3.** Molecular structure of **7**. The OLEX2 drawing, with a 20% probability thermal ellipsoid, shows the atom-labeling. Important bond lengths (Å) and angles (°): Ag1-Cl1 2.3256(8), Ag1-C1 2.085(3), N1-C1 1.353(3), N1-C2, 1.390(3), N1-C8 1.459(3), N2-C1 1.345(3), N2-C7 1.387(3), N2-C12 1.475(3), C2-C7 1.384(4), Cl1-Ag1-C1 174.81(7), Ag1-C1-N1 125.40(18), C1-N1-C2 111.3(2), N1-C2-C7 105.7(2), C2-C7-N2 106.1 (2), C1-N2-C7 111.4(2), N2-C1-N1 105.4(2), N2-C1-Ag1 129.10(18), C1-N1-C8 124.8(2), C2-N1-C8 123.9(2), C1-N2-C12 125.5(2), and C7-N2-C12 123.0(2).

The silver atom is almost linearly coordinated to the carbonic carbon and to the chloride with a pitch angle of 5.2° (C1-Ag1-Cl1 angle of 174.81(7)°) [45]. The Ag1-Cl1 moiety is oriented on one side of the benzyl ring, resulting in two different Ag1-C15 (4.348 Å) and Ag1-C22 (3.561 Å) lengths. The tetramethylphenyl ring is almost orthogonal to the coordinated benzimidazolydene (dihedral angle of 85.77°) as observed in the related chloro[1-methallyl-3-(2,3,4,5,6-pentamethylbenzyl)-5,6-dimethylbenzimidazol-2-ylidene]silver(I) complex [46]. The Ag1-Cl1 and Ag1-C1 distances (2.3256(8) and 2.085(3) Å, respectively) and the N1-C1-N2 angle (105.4(2)°) are within the range usually observed for (NHC)Ag–Cl complexes [47–51]. We can observe that the two N1-C1 and N2-C1 bonds (1.353(3) and 1.345(3) Å, respectively) are shorter than the two N1-C2 and N2-C7 bonds (1.390(3) and 1.387(3) Å, respectively); this difference could be explained by delocalization within the imidazole ring. Furthermore, the tetramethylphenyl and 2-methylallyl moieties are in syn positions with respect to the carbenic ring.

## 2.3. Antimicrobial and Antibiofilm Activities

Benzimidazolium salts **1–5** and their silver metal complexes **6–10** were tested as antimicrobial and antibiofilm agents with microorganisms, which commonly cause nosocomial infections [52], namely *Staphylococcus aureus*, *Enterococcus faecalis* (two Gram-positive bacteria), *Escherichia coli*, *Acinetobacter baumannii* (two Gram-negative bacteria) and *Candida albicans*. These strains were obtained from the American Type Culture Collection (ATCC).

As shown in Table 2, the tested benzimidazolium salts and silver complexes displayed antimicrobial activity at certain concentrations. Good minimum inhibitory concentrations (MIC) were obtained for the salts **1–5** for the selected microorganisms with values in the range from 7.8 to 125.0 μg/mL, except for the Gram-positive *Staphylococcus aureus*, for

which the MIC values are strongly dependent on the nature of the aromatic substituent on the benzimidazole ring and vary between moderate 125.0 µg/mL for salt **1** (benzyl) to excellent 7.8 µg/mL for salt **5** (anthracen-9-yl-methyl) concentrations. The introduction of a silver atom has a beneficial effect on antibacterial activities. With silver complexes **6–10**, the maximum MIC value measured for the five microorganism strains was 15.6 µg/mL. The complex bearing the anthracen-9-yl-methyl substituent **10** proved to be the most effective, with MIC values as low as 1.9 µg/mL against *Staphylococcus aureus* and *Candida albicans* strains. These MIC values are close to those obtained with commonly used Ciprofloxacin antibiotic and Fluconazole antifungal or silver complexes based on multi-donor heterocyclic thioamides for which bis-silver drugs could lead to MIC in the range 1.9–15.6 µg/mL [53].

**Table 2.** Minimum inhibitory concentrations (MICs) of compounds (µg/mL).

| | Microorganism | | | | |
| Drug | *Staphylococcus aureus* | *Enterococcus faecalis* | *Escherichia coli* | *Acinetobacter baumannii* | *Candida albicans* |
|---|---|---|---|---|---|
| **1** | 125.0 | 62.5 | 62.5 | 62.5 | 62.5 |
| **2** | 31.2 | 62.5 | 62.5 | 62.5 | 62.5 |
| **3** | 125.0 | 62.5 | 62.5 | 62.5 | 62.5 |
| **4** | 15.6 | 62.5 | 62.5 | 62.5 | 31.2 |
| **5** | 7.8 | 31.2 | 62.5 | 62.5 | 31.2 |
| **6** | 15.6 | 15.6 | 7.8 | 7.8 | 7.8 |
| **7** | 15.6 | 15.6 | 7.8 | 7.8 | 7.8 |
| **8** | 7.8 | 7.8 | 3.9 | 7.8 | 7.8 |
| **9** | 7.8 | 7.8 | 3.9 | 7.8 | 7.8 |
| **10** | 1.9 | 7.8 | 3.9 | 7.8 | 1.9 |
| Ciprofloxacin | <1.9 | <1.9 | <1.9 | <1.9 | / |
| Fluconazole | / | / | / | / | <1.9 |

The good performances of compounds **1–5** and **6–10** as antimicrobial agents allowed us to study their ability to reduce the more challenging biofilm formation [54]. The antibiofilm activities were determined at 0.5 MIC concentrations (Table 3). The five benzimidazolium salts **1–5** displayed moderate antibiofilm activities towards *Enterococcus faecalis*, *Escherichia coli*, *Acinetobacter baumannii*, and *Candida albicans*, with biofilm inhibition of 37.6–63.4% with concentrations of 32.2 or 15.6 µg/mL. The higher activities were obtained using *Staphylococcus aureus* and salt **5**, for which a biofilm inhibition of 71.8% was measured at 3.9 µg/mL. As an attempt, the antibiofilm effect of silver complexes was more significant with a minimum of biofilm inhibition of 76.1 ± 0.1 at a similar concentration (3.9 µg/mL) measured using the silver complex **8**. The higher biofilm inhibition, 92.9% at 1.9 µg/mL, was obtained for the Gram-negative *Escherichia coli* using silver complex **10**. Note that for technical reasons, with the latter complex, sub-MIC concentration could not be tested. Nevertheless, at MIC concentration (1.9 µg/mL of **10**), biofilm inhibition higher than 90% was measured for the two microorganism strains, *Staphylococcus aureus* and *Candida albicans*.

### 2.4. Molecular Docking

Deoxyribonucleic acid (DNA) gyrase is a topoisomerase-type enzyme in prokaryotic cells and has a critical role in DNA replication. The enzyme puts the negative supercoils during replication and ensures the proceeding of the processes [55]. Therefore, DNA gyrase could be a convenient target for disrupting DNA replication and other vital cellular processes, leading to bacterial death [56]. Moreover, DNA gyrase can be accepted as a key object for the treatment of bacterial infections and the design of new antibacterials. For instance, the primary action mechanism of Ciprofloxacin, which was used for ranking in vitro experiments in this study, is the inhibition of bacterial DNA gyrase [57]. Consequently, DNA gyrase was a target molecule in molecular docking studies examining antibacterial activity; for example, Patil and co-workers analyzed the potential utilization

of new piperazine derivatives using the molecular docking method against DNA gyrase and recorded consensus scores in the range of 5.55–2.41 kcal/mol, which are accepted as very good docking scores against *Escherichia coli* [58]. Düşünceli and co-workers performed molecular docking against DNA gyrase to provide insights into the antibacterial activity of benzimidazol-2-ylidene silver(I) complexes and recorded remarkable interactions with the region, including Met91, Val120, Leu130, and Val167, with a binding energy of 8 kcal/mol [59]. Anastasiadou and co-workers studied the silver complexes with thioamide and arylphosphane type ligands against DNA gyrase and recorded the remarkable interactions with the protein's binding pocket, including Asn46, Glu50, Ile78, Ile90, Val120, and Thr165 residues [60].

**Table 3.** Biofilm inhibition (%) measured at 0.5 MIC (µg/mL).

| | Microorganism | | | | |
| Drug | *Staphylococcus aureus* | *Enterococcus faecalis* | *Escherichia coli* | *Acinetobacter baumannii* | *Candida albicans* |
|---|---|---|---|---|---|
| **1** | 62.5 µg/mL 36.5 ± 1.1% | 31.2 µg/mL 48.5 ± 0.3% | 31.2 µg/mL 59.7 ± 0.6% | 31.2 µg/mL 46.5 ± 0.7% | 31.2 µg/mL 40.9 ± 0.5% |
| **2** | 15.6 µg/mL 47.7 ± 0.4% | 31.2 µg/mL 44.6 ± 0.8% | 31.2 µg/mL 63.4 ± 0.5% | 31.2 µg/mL 49.2 ± 0.8% | 31.2 µg/mL 43.4 ± 0.4% |
| **3** | 62.5 µg/mL 43.2 ± 0.6% | 31.2 µg/mL 50.2 ± 0.1% | 31.2 µg/mL 54.8 ± 1.0% | 31.2 µg/mL 57.3 ± 0.2% | 31.2 µg/mL 7.6 ± 0.7% |
| **4** | 7.8 µg/mL 57.9 ± 0.4% | 31.2 µg/mL 42.7 ± 0.2% | 31.2 µg/mL 59.3 ± 0.4% | 31.2 µg/mL 58.0 ± 0.3% | 15.6 µg/mL 45.6 ± 0.2% |
| **5** | 3.9 µg/mL 71.8 ± 0.3% | 15.6 µg/mL 55.4 ± 0.3% | 31.2 µg/mL 62.6 ± 0.3% | 31.2 µg/mL 55.8 ± 0.3% | 15.6 µg/mL 50.8 ± 0.2% |
| **6** | 7.8 µg/mL 61.6 ±0.4% | 7.8 µg/mL 54.6 ± 0.1% | 3.9 µg/mL 75.9 ± 0.3% | 3.9 µg/mL 77.1 ± 0.7% | 3.9 µg/mL 81.2 ± 0.2% |
| **7** | 7.8 µg/mL 68.4 ± 0.3% | 7.8 µg/mL 59.0 ± 0.1% | 3.9 µg/mL 86.6 ± 0.2% | 3.9 µg/mL 73.9 ± 0.4% | 3.9 µg/mL 78.0 ± 0.3% |
| **8** | 3.9 µg/mL 76.1 ± 0.1% | 3.9 µg/mL 63.3 ± 0.2 | 1.9 µg/mL 90.1 ± 0.1 | 3.9 µg/mL 83.2 ± 0.2% | 3.9 µg/mL 76.5 ± 0.3% |
| **9** | 3.9 µg/mL 67.3 ± 0.2% | 3.9 µg/mL 59.6 ± 0.1% | 1.9 µg/mL 86.5 ± 0.3% | 3.9 µg/mL 80.7 ± 0.1% | 3.9 µg/mL 84.3 ± 0.3% |
| **10** | Not tested | 3.9 µg/mL 70.8 ± 0.4% | 1.9 µg/mL 92.9 ± 0.2% | 3.9 µg/mL 84.0 ± 0.2% | Not tested |

The interactions of the five silver complexes **6**–**10** against DNA gyrase were investigated using molecular docking method, and the results were compared to that of Ciprofloxacin (Table 4). All molecules interacted with the same region of the macromolecule, and H-bonds were observed with all silver drugs (Figure 4). The highest interaction was measured for complex **9** with a binding affinity of −8.11 kcal/mol and came mainly from a strong H-bond (3.11 Å) with Asn46 and π-interactions with Glu50 and Thr165. On the other hand, complex **10** displayed a strong H-bond (2.69 Å) with the carboxy region of Asn46, as well as a weak H-bond (3.39 Å) through the nitrogen in the amino region for a −8.01 kcal/mol binding constant. Conversely, despite the H-bonds with Asn46 (2.83 Å, strong), Asn49 (2.91 Å, strong), and Thr165 (3.37 Å, weak), complex **8** presented the weakest binding affinity (−6.21 kcal/mol) [61]. The binding affinities of complexes **6** and **7** were calculated as −7.26 and −7.50 kcal/mol, respectively. The weak H-bond of **6** with Thr165 has a 3.23 Å bond length, while the weak H-bond of **7** with Asn45 has a 3.44 Å bond length. All molecules except **8** have higher binding affinity than Ciprofloxacin.

**Table 4.** Active sites of the crystal structure of *Escherichia coli* DNA gyrase and CYP51 from the pathogen *Candida albicans* with silver complexes **6–10**.

| Drugs | Binding Energy (kcal/mol) | Amino Acids Residue [1] |
|:---:|:---:|:---:|
| *Escherichia coli* DNA gyrase | | |
| 6 | −7.26 | Thr165, Glu50, Val43, Val47, Arg76, Ile78, Pro79, Ile90, Val167, . . . |
| 7 | −7.50 | Asn45, Thr165, Ala47, Arg76, Ile78, Pro79, Ile90, Val120, . . . |
| 8 | −6.21 | Asn46, Asp49, Thr165, Glu50, Arg76, Gly77, Pro79, Ile78, . . . |
| 9 | −8.11 | Asn46, Glu50, Thr165, Ala47, Ile78, Ile90, Met91, Val120, Val167 |
| 10 | −8.01 | Asn46, Glu50, Ile78, Met91, Val120, Thr165, Ala47, Val167, Val43 |
| Ciprofloxacin | −6.89 | Asp73, Arg76, Arg136, Thr165, Glu50, Gly77, Ile78, Pro79, . . . |
| CYP51 from the pathogen *Candida albicans* | | |
| 6 | −7.94 | Gly303, Ile131, Leu139, Lys143, Ala146, Leu150, Leu300, Ile304, Phe126, . . . |
| 7 | −8.25 | Ile131, Leu139, Lys143, Ala146, Leu150, Leu204, Leu276, Leu300, Ile304, . . . |
| 8 | −8.52 | Cys470, Gly472, Ile304, Ile471, Ile131, Tyr132, Leu139, Lys143, . . . |
| 9 | −8.51 | Leu150, Ile304, Ile131, Ala146, Leu204, Leu276, Leu300, Cys470, Ile471, . . . |
| 10 | −9.26 | Ile131, Leu300, Lys143, Ala146, Leu150, Leu204, Leu276, Ile304, Cys470, . . . |
| Fluconazole | −5.65 | Ile304, Gly307, Cys470, Gly303, Ile471, Ile131, Lys143, Leu204, . . . |

[1] red: H-bond, orange: pi-interactions, blue: alkylic interactions, green: van der Waals interactions, and purple: halogenic interactions.

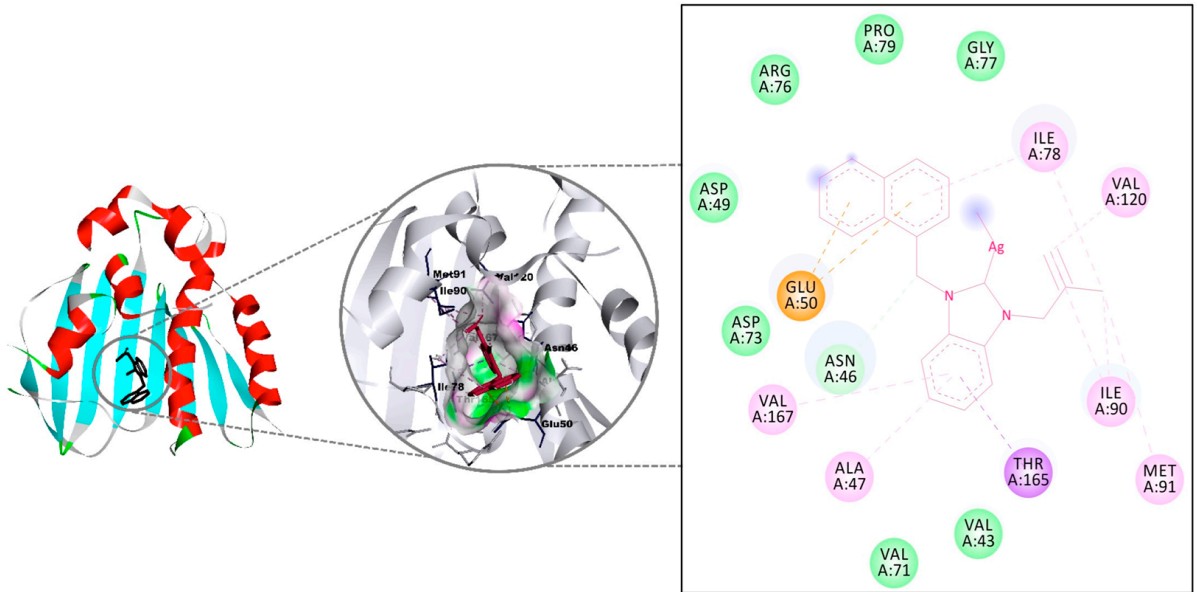

**Figure 4.** The interaction details of complex **9** with DNA gyrase of *Escherichia coli*.

Lanosterol 14α-demethylase (CYP51) is an enzyme of the cytochrome P450 family that plays an essential role in the biosynthesis of sterols, which are key components of the cell membranes of fungi. Fungal cell membranes become more weakened and permeable with an insufficient level of ergosterol, which results in cell death [62]. The molecules, able

to inhibit CYP51, are important for developing the antifungal agents and understanding the antifungal resistance mechanisms. Fluconazole, which was used as a ranking drug in this study, is one of the antifungal molecules able to interact with CYP51 and impair cell membrane permeability [63]. Therefore, CYP51 is the target molecule of choice in many molecular docking studies examining antifungal activity. As example, Kartsev and co-workers analyzed the probable mechanism of the antifungal activity of heteroaryl(aryl) thiazole-type molecules with molecular docking method against CYP51 and recorded the remarkable interactions with Tyr64, Tyr118, and Tyr132 residues [64]. Ismael and co-workers performed the docking to understand the interaction of bidentate Schiff base ligands and their metal complexes against the 14α-demethylase enzyme and achieved the interactions with His374, Ser375, His377, and Gln66 with good binding affinity values [65]. Souza and co-workers analyzed a family of zinc(II) complexes bearing polydentate-*N,N,S* ligands against the CYP51 enzyme. The calculations revealed that one complex displayed a docking score of -11.12 kcal/mol, mainly due to hydrogen bonds with Tyr118 and Hem601 and confirmation of its considerable antifungal activity [66].

The interactions of complexes **6**–**10** were investigated by the molecular docking method, and the results were compared to those of Fluconazole (Table 4). The all complexes interacted with the same region of the macromolecule (Figure 5). The highest interaction was recorded for complex **10**, which was in agreement with the results recorded in vitro experiments for *Candida albicans*. Two H-bonds were observed with Gly472 (2.99 Å, strong) and Cys470 (3.90 Å, weak) for complex **8**, and the binding constant of −8.52 kcal/mol was determined as a result of these interactions. The binding values recorded for complexes **6**, **7**, and **9** were calculated as −7.94, −8.25, and −8.51 kcal/mol, respectively. All binding energy calculated for the five silver complexes was lower than the calculated value for Fluconazole (5.65 kcal/mol) (see Supplementary Materials).

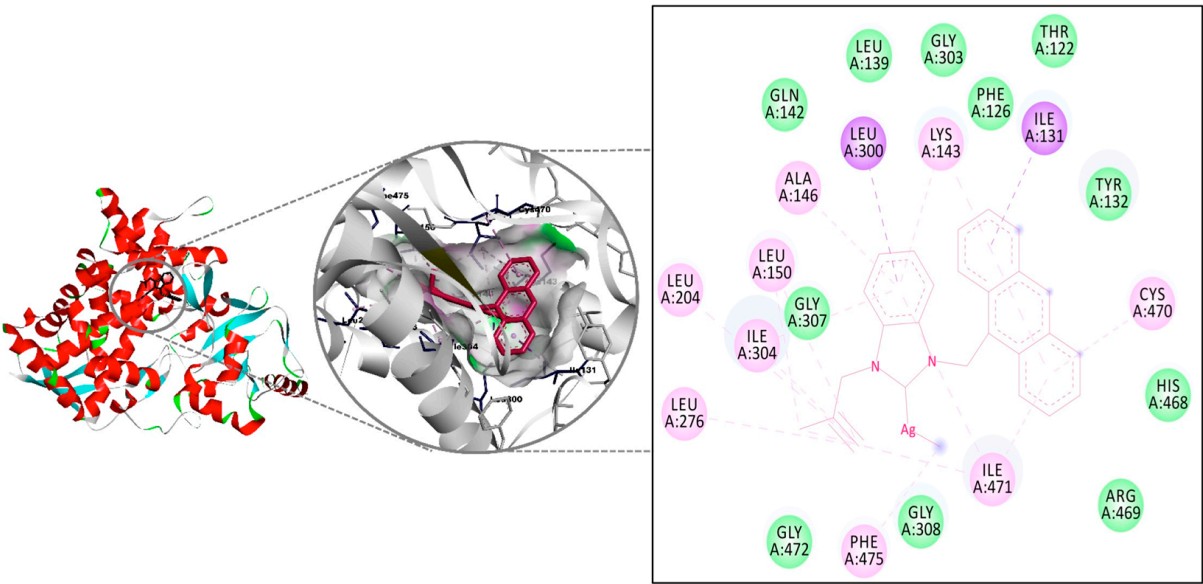

**Figure 5.** The interaction details of complex **10** with CYP51 of *Candida albicans*.

The highest occupied molecular orbitals (HOMO) and the lowest unoccupied molecular orbitals (LUMO) are the most reactive regions of molecules. These orbitals give useful insight into biological and chemical activity. LUMO behaves as a Lewis base, while the HOMO region acts as a Lewis acid [67]. In the present study, HOMOs of complexes **6**–**10** were located on the chloride atom while LUMOs of complexes **6**, **7,** and **8** were located on the benzimidazole ring and for complexes **9** and **10** on naphthyl and anthracenyl substituents (Figure 6). In general, ionization potential (IP), electron affinity (EA), electronegativity (χ), chemical hardness (η), global softness (S), and the electrophilicity index (ω) are considerable criteria for having an idea about the structure–activity relationship

(Table 5). The HOMO-LUMO Gap (eV), which is important for intramolecular charge transfer, order of the molecules was recorded as **10** (1.731 eV) < **9** (2.314 eV) < **8** (2.452 eV) < **6** (2.474 eV) < **7** (2.534 eV). The electron affinity values of the molecules were recorded as **8** (2.283 eV) < **7** (2.297 eV) < **6** (2.379 eV) < **9** (2.495 eV) < **10** (3.126 eV). Consequently, complex **10** with the lowest HOMO-LUMO gap should be considered to be a softer and more reactive compound due to lower kinetic stability and higher polarity.

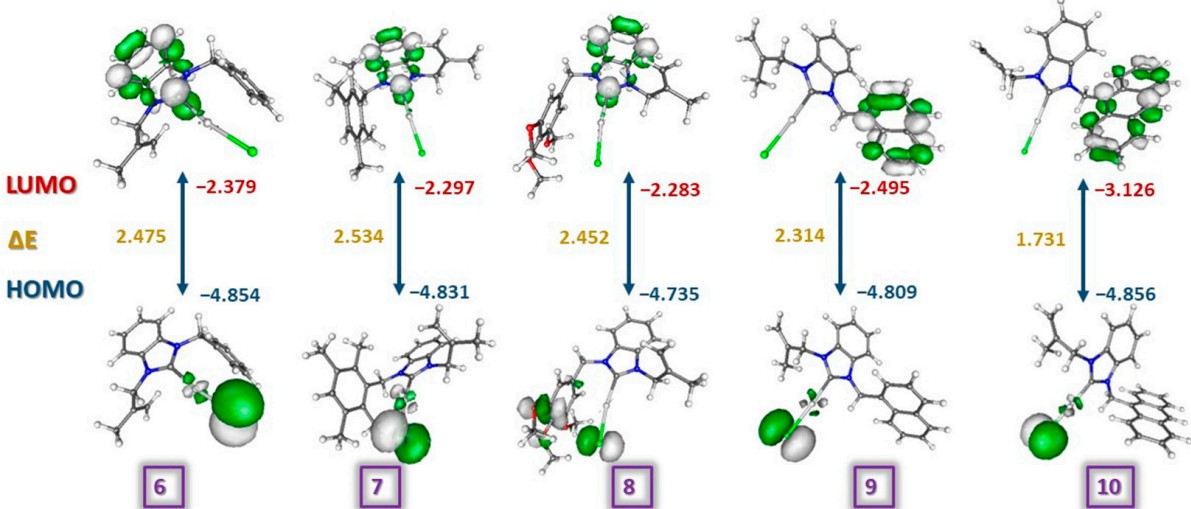

**Figure 6.** The illustrations and the frontier orbital energies of complexes **6–10**.

**Table 5.** Spectroscopic data of benzimidazolium salts **1–5** and their corresponding silver complexes **6–10**.

|  | 6 | 7 | 8 | 9 | 10 |
|---|---|---|---|---|---|
| HOMO-LUMO Gap ($\Delta E$) | 2.474 | 2.534 | 2.452 | 2.314 | 1.731 |
| Ionization Potential (IP) | 4.854 | 4.831 | 4.735 | 4.809 | 4.856 |
| Electron Affinity (EA) | 2.379 | 2.297 | 2.283 | 2.495 | 3.126 |
| Electronegativity ($\chi$) | 3.616 | 3.564 | 3.509 | 3.652 | 3.991 |
| Chemical Hardness ($\eta$) | 1.237 | 1.267 | 1.226 | 1.157 | 0.865 |
| Global Softness (S) | 0.404 | 0.395 | 0.408 | 0.432 | 0.578 |

## 3. Materials and Methods

The syntheses of the silver complexes were made under an inert atmosphere (Ar) using standard Schlenk techniques. All reagents were supplied by Sigma-Aldrich Chemical Co and Merck. The Electrothermal-9200 melting point apparatus was used for the melting point measurements under air in glass capillaries, and the temperatures were reported as uncorrected values. FT-IR spectra were generated with a Perkin Elmer 100 spectrometer. $^{1}$H and $^{13}$C{$^{1}$H} NMR spectra were recorded with a 400 MHz Bruker Avance III spectrometer. $^{1}$H NMR and $^{13}$C{$^{1}$H} spectra were referenced to residual protonated solvents ($\delta$ = 7.26 ppm and 77.16 ppm for $CDCl_3$, respectively, and 2.50 ppm and 39.52 ppm for $(CD_3)_2SO$, respectively). An Agilent Technologies LC/MSD SL mass spectrometer was employed with nitrogen as a nebulizer and dry gas. LC-MS samples were obtained at room temperature with a continuous flow of 1 mL/min using a $C_{18}$ column (250/4.6 mm). 1-Methallyl-3-benzylbenzimidazolium chloride (**1**), 1-methallyl-3-(2,3,5,6-tetramethyl-benzyl)benzimidazolium chloride (**2**), 1-methallyl-3-(3,4,5-trimethoxy-benzyl)benzimidazolium chloride (**3**), 1-methallyl-3-(naphthylmethyl)benzimidazolium chloride (**4**), and 1-methallyl-3-(anthracen-9-yl-methyl)benzimidazolium chloride (**5**) were prepared by following literature procedures [36–38].

### 3.1. General Procedure for the Synthesis of Silver Complexes

Under an argon atmosphere, a solution of benzimidazolium salt (1.0 mmol) in dichloromethane (50 mL) was prepared, and silver(I)oxide (2.2 mmol) was added and stirred. After 24 h at room temperature and in the dark, the mixture was filtered through a celite bed. The resulting clear solution was concentrated under vacuum to ca 5 mL before the addition of diethyl ether (60 mL). The precipitated complexes **6–10** were recovered by filtration, washed (diethyl ether), and dried under vacuum.

Chloro[1-methallyl-3-benzyl)benzimidazol-2-ylidene]silver(I) (**6**) Yield: 79%; m.p. 215–216 °C; FT-IR $\nu$(CN): 1397 cm$^{-1}$; LC-MS: 631.3 [AgL$_2$ − H]$^+$, 671.3 [AgL$_2$ + K]$^+$; $^1$H NMR (400 MHz, (CD$_3$)$_2$SO) $\delta$ = 1.69 (s, 3H, NCH$_2$C(C*H$_3$*)CH$_2$), 4.76 (s, 1H, NCH$_2$C(CH$_3$)C*H$_2$*), 4.96 (s, 1H NCH$_2$C(CH$_3$)C*H$_2$*), 5.09 (s, 2H, NC*H$_2$*C(CH$_3$)CH$_2$), 5.75 (s, 2H, C*H$_2$*C$_6$H$_5$), 7.28–7.37 (m, 5H, CH arom.), 7.38–7.43 (m, 2H, CH arom.), 7.70–7.75 (m, 2H, CH arom.); $^{13}$C{$^1$H} NMR (100 MHz, (CD$_3$)$_2$SO) $\delta$: 19.87 (s, NCH$_2$C(*C*H$_3$)CH$_2$), 51.90 (s, *C*H$_2$C$_6$H$_5$), 53.96 (s, N*C*H$_2$C(CH$_3$)CH$_2$), 124.09 (s, NCH$_2$*C*(CH$_3$)CH$_2$), 140.49 (s, NCH$_2$C(CH$_3$)*C*H$_2$), 112.39, 113.40, 127.20, 127.33, 128.02, 128.80, 133.15, 133.68, 136.33 (9s, arom Cs), 189.27 (C$_{carbene}$-Ag) ppm.

Chlro[1-methallyl-3-(2,3,5,6-tetramethylbenzyl)benzimidazol-2-ylidene]sil-ver(I) (**7**) Yield: 81%; m.p. 189–190 °C; FT-IR $\nu$(CN): 1396 cm$^{-1}$; LC-MS: 745.4 [AgL$_2$ + H]$^+$; $^1$H NMR (400 MHz, CDCl$_3$) $\delta$ = 1.67 (s, 3H, NCH$_2$C(C*H$_3$*)CH$_2$), 2.14 (s, 6H, C$_6$H(C*H$_3$*)$_4$), 2.29 (s, 6H, C$_6$H(C*H$_3$*)$_4$), 4.75 (s, 1H, NCH$_2$C(CH$_3$)C*H$_2$*), 4.92 (s, 2H, NC*H$_2$*C(CH$_3$)CH$_2$), 4.97 (s, 1H, NCH$_2$C(CH$_3$)C*H$_2$*), 5.50 (s, 2H, C*H$_2$*C$_6$H(CH$_3$)$_4$), 7.14 (s, 1H, CH arom of C$_6$*H*(CH$_3$)$_4$), 7.35–7.46 (m, 4H, CH arom of C$_7$*H$_4$*N$_2$); $^{13}$C{$^1$H} NMR (100 MHz, CDCl$_3$) $\delta$ = 16.32 (s, C$_6$H(*C*H$_3$)$_4$), 20.11 (s, NCH$_2$C(*C*H$_3$)CH$_2$), 20.83 (s, C$_6$H(*C*H$_3$)$_4$), 47.71 (s, *C*H$_2$C$_6$H(CH$_3$)$_4$), 56.42 (s, N*C*H$_2$C(CH$_3$)CH$_2$), 112.23 (s, NCH$_2$*C*(CH$_3$)CH$_2$), 139,22 (s, NCH$_2$C(CH$_3$)*C*H$_2$), 111.47, 114.39, 124.30, 124.48, 129.74, 133.33, 133.59, 133.95, 134.52, 135.50 (10s, arom Cs), 187.82 (C$_{carbene}$-Ag) ppm.

Chloro[1-methallyl-3-(3,4,5-trimethoxylbenzyl)benzimidazol-2-ylidene]silver(I) (**8**) Yield: 77%; m.p. 239–240 °C; FT-IR $\nu$(CN): 1392 cm$^{-1}$; LC-MS: 813.3 [AgL$_2$ + H]$^+$; $^1$H NMR (400 MHz, (CD$_3$)$_2$SO) $\delta$ = 1.69 (s, 3H, NCH$_2$C(C*H$_3$*)CH$_2$), 3.61 (s, 3H, C$_6$H$_2$(OC*H$_3$*)$_3$), 3.70 (s, 6H, C$_6$H$_2$(OC*H$_3$*)$_3$), 4.73 (s, 1H, NCH$_2$C(CH$_3$)C*H$_2$*), 4.95 (s, 1H, NCH$_2$C(CH$_3$)C*H$_2$*), 5.08 (s, 2H, NC*H$_2$*C(CH$_3$)CH$_2$), 5.63 (s, 2H, C*H$_2$*C$_6$H$_2$(OCH$_3$)$_3$), 6.77 (s, 2H, CH arom of C$_6$*H$_2$*(OCH$_3$)$_3$), 7.41–7.43 (m, 2H, CH arom of C$_7$*H$_4$*N$_2$), 7.72–7.74 (m, 1H, CH arom of C$_7$*H$_4$*N$_2$), 7.86–7.88 (m, 1H, CH arom of C$_7$*H$_4$*N$_2$); $^{13}$C{$^1$H} NMR (100 MHz, (CD$_3$)$_2$SO) $\delta$ = 19.82 (s, NCH$_2$C(*C*H$_3$)CH$_2$), 51.87 (s, *C*H$_2$C$_6$H$_2$(OCH$_3$)$_3$), 54.00 (s, N*C*H$_2$C(CH$_3$)CH$_2$), 55.87 (s, C$_6$H$_2$(O*C*H$_3$)$_3$), 59.98 (s, C$_6$H$_2$(O*C*H$_3$)$_3$), 124.13 (s, NCH$_2$*C*(CH$_3$)CH$_2$), 140.56 (s, NCH$_2$C(CH$_3$)*C*H$_2$), 105.07, 112.37, 113.22, 124.10, 131.82, 133.26, 133.55, 137.15, 153.04 (9s, arom Cs), 189.03 (C$_{carbene}$-Ag) ppm.

Chloro[1-methallyl-3-(naphthylmethyl)benzimidazol-2-ylidene]silver(I) (**9**) Yield: 74%; m.p. 145–146 °C; FT-IR $\nu$(CN): 1392 cm$^{-1}$, LC-MS: 731.3 [AgL$_2$ − H]$^+$; $^1$H NMR (400 MHz, CDCl$_3$) $\delta$ = 1.74 (s, 3H, NCH$_2$C(C*H$_3$*)CH$_2$), 4.86 (s, 1H, NCH$_2$C(CH$_3$)C*H$_2$*), 5.02 (s, 2H, NC*H$_2$*C(CH$_3$)CH$_2$), 5.04 (s, 1H, NCH$_2$C(CH$_3$)C*H$_2$*), 6.08 (s, 2H, C*H$_2$*C$_{10}$H$_7$), 6.97 (d, 1H, $^3J_{HH}$ = 7.2 Hz, CH arom of C$_{10}$*H$_7$*), 7.22–7.28 (m, 1H, CH arom of C$_{10}$*H$_7$*), 7.33–7.37 (m, 2H, CH arom of C$_7$*H$_4$*N$_2$), 7.48 (d, 1H, $^3J_{HH}$ = 8.4 Hz, CH arom of C$_{10}$*H$_7$*), 7.52–7.58 (m, 2H, CH arom of C$_{10}$*H$_7$* and C$_7$*H$_4$*N$_2$), 7.72–7.80 (m, 1H, CH arom of C$_7$*H$_4$*N$_2$), 7.83 (d, 1H, $^3J_{HH}$ = 8.0 Hz, CH arom of C$_{10}$*H$_7$*), 7.89–7.91 (m, 1H, CH arom of C$_{10}$*H$_7$*), 7.99–8.01 (m, 1H, CH arom of C$_{10}$*H$_7$*); $^{13}$C{$^1$H} NMR (100 MHz, CDCl$_3$) $\delta$ = 20.16 (s, NCH$_2$C(*C*H$_3$)CH$_2$), 51.45 (s, *C*H$_2$C$_{10}$H$_7$), 55.85 (s, N*C*H$_2$C(CH$_3$)CH$_2$), 112.26 (s, NCH$_2$*C*(CH$_3$)CH$_2$), 139.18 (s, NCH$_2$C(CH$_3$)*C*H$_2$), 112.31, 114.78, 122.41, 124.52, 124.55, 125.02, 125.45, 126.46, 127.15, 129.32, 129.41, 130.16, 130.63, 133.96, 134.14, 134.19 (16s, arom Cs), no peak (C$_{carbene}$-Ag) ppm.

Chloro[1-methallyl-3-(anthracen-9-yl-methyl)benzimidazol-2-ylidene]silver(I) (**10**) Yield: 71%; m.p. 277–278 °C; FT-IR $\nu$(CN): 1394 cm$^{-1}$; LC-MS: 833.4 [AgL$_2$ + H]$^+$; $^1$H NMR (400 MHz, CDCl$_3$) $\delta$ = 1.66 (s, 3H, NCH$_2$C(C*H$_3$*)CH$_2$), 4.74 (s, 1H, NCH$_2$C(CH$_3$)C*H$_2$*), 4.91 (s, 2H, NC*H$_2$*C(CH$_3$)CH$_2$), 4.96 (s, 1H, NCH$_2$C(CH$_3$)C*H$_2$*), 6.47 (s, 2H, C*H$_2$*C$_{14}$H$_9$),

7.02 (d, 1H, $^3J_{HH}$ = 8.4 Hz, CH arom of $C_7H_4N_2$), 7.09 (t, 1H, $^3J_{HH}$ = 7.8 Hz, CH arom of $C_{14}H_9$), 7.25 (t, 1H, $^3J_{HH}$ = 7.8 Hz, CH arom of $C_{14}H_9$), 7.37 (d, 1H, $^3J_{HH}$ = 8.4 Hz, CH arom of $C_7H_4N_2$), 7.48–7.56 (m, 4H, CH arom of $C_{14}H_9$ and $C_7H_4N_2$), 8.09 (d, 2H, $^3J_{HH}$ = 8.0 Hz, CH arom of $C_{14}H_9$), 8.22 (d, 2H, $^3J_{HH}$ = 8.8 Hz, CH arom of $C_{14}H_9$), 8.61 (s, 1H, CH arom of $C_{14}H_9$); $^{13}$C{$^1$H} NMR (100 MHz, CDCl$_3$) $\delta$ = 20.10 (s, NCH$_2$C(CH$_3$)CH$_2$), 49.99 (s, $CH_2C_{14}H_9$), 56.06 (s, NCH$_2$C(CH$_3$)CH$_2$), 112.21 (s, NCH$_2$C(CH$_3$)CH$_2$), 139.17 (s, NCH$_2$C(CH$_3$)CH$_2$), 112.11, 114.58, 123.00, 123.16, 123.30, 124.34, 125.47, 127.81, 130.17, 130.51, 131.23, 131.57, 134.17, 134.22 (14s, arom Cs), no peak ($C_{carbene}$-Ag) ppm.

### 3.2. X-ray Crystallography

Slow diffusion of diethylether into a dichloromethane solution of complex **7** led to the formation of single crystals suitable for X-ray analysis. The measurements were performed on an STOE IPDS II diffractometer using Mo K$\alpha$ radiation and $\omega$-scans at 296(2) K. All data collection, cell refinement, data reduction, and correction procedures were performed using the X-AREA and X-RED32 software [68]. The structure was solved by direct methods with SIR2019 [69] and refined against F2 using the full-matrix least-squares method with the SHELXL-2019 program [70]. H atoms were included in idealized positions and constrained to ride on their parent atoms. A summary of the key crystallographic information is collected in Table 6. OLEX2 [71] was employed for the generation of the molecular graphic. CCDC entry 2018964 contains the supplementary crystallographic data for **7**. These data can be downloaded from The Cambridge Crystallographic Data Centre via www.ccdc.cam.ac.uk/structures, accessed on 20 February 2023.

**Table 6.** Crystal data and structure refinement parameters for complex **7**.

| | | | |
|---|---|---|---|
| CCDC Depository | 2018964 | Color/Shape | Colorless/Prism |
| Chemical formula | AgClC$_{22}$H$_{26}$N$_2$ | Formula weight | 461.77 |
| Temperature (K) | 296(2) | Wavelength (Å) | 0.71073 Mo K$\alpha$ |
| Crystal system | Monoclinic | Space group | C2/c |
| Unit cell parameters | | Z | 8 |
| $a$ (Å) | 28.1169(14) | $D_{calc.}$ (g/cm$^3$) | 1.432 |
| $b$ (Å) | 9.3573(6) | $\mu$ (mm$^{-1}$) | 1.073 |
| $c$ (Å) | 17.0595(8) | $\theta$ range for data collection (°) | $2.305 \le \theta \le 27.570$ |
| $\alpha$ (°) | 90 | Absorption correction | Integration |
| $\beta$ (°) | 107.331(4) | F000 | 1888 |
| $\gamma$ (°) | 90 | Diffractometer | STOE IPDS II |
| Index ranges | $-33 \le h \le 36$ | Reflections collected | 14053 |
| | $-10 \le k \le 12$ | $R_{int.}$ | 0.0365 |
| | $-22 \le l \le 22$ | Data/restraints/parameters | 4899/1/235 |
| Independent reflections | 4899 | Observed reflections | 3493 |
| Refinement method | Full-matrix least-squares on $F^2$ | Final $R$ indices [$I > 2\sigma(I)$] | $R_1$ = 0.0327, $wR_2$ = 0.0788 |
| Goodness-of-fit on $F^2$ | 0.918 | $R$ indices (all data) | $R_1$ = 0.0523, $wR_2$ = 0.0842 |
| $\Delta\rho_{max.}$, $\Delta\rho_{min.}$ (e/Å$^3$) | 0.603, $-0.410$ | | |

### 3.3. Biological Assays

#### 3.3.1. Antimicrobial Activity

Per the advice of international standards, the antibacterial activities of the compounds were determined using the liquid microdilution method [72]. The method was used to investigate the minimum inhibitory concentration (MIC) values of the silver compounds for the yeast and bacterial strains. Standard bacterial strains *Staphylococcus aureus* ATCC 29213, *Enterococcus faecalis* ATCC 29212, *Escherichia coli* ATCC 25922, *Acinetobacter baumannii* ATCC 17978, and yeast *Candida albicans* ATCC 10231, were previously stored at $-80$ °C then revived at room temperature. Microorganisms were inoculated using overnight

broth cultures, and suspensions ($1 \times 10^8$ CFU/mL microorganisms) were maintained at 0.5 McFarland standard turbidity. Compounds weighing 2 mg were diluted in 4 mL of Mueller–Hinton Broth (MHB) to form a stock solution. All compounds to be tested were dissolved in MHB containing 10% (*v/v*) DMSO, followed by sequential two-fold dilutions in 96-well plates with MHB at concentrations ranging from 1.9 to 125 g/mL. Aliquots (100 μL) were dispensed into the wells of the MHB 96 well microplate. Initial concentrations of the compounds were distributed on them. Then, two-fold dilutions were made sequentially. The microorganisms were suspended in 100 μL of liquid and added to each test well. Ciprofloxacin and Fluconazole were used for the positive control in the study. Microorganism suspensions (200 μL) were used as negative controls. The optical density was measured at 620 nm. The first concentration without microbial growth was determined as MIC.

### 3.3.2. Reduction in Biofilm Formation

The well plate assay was utilized to test the effectiveness of compounds in inhibiting biofilms made by microorganisms. Microorganisms in TSB (Tryptic Soy Broth) containing 1% glucose were adjusted at a density of $1 \times 10^8$ CFU/mL. Then, 100 μL of these suspensions were dispensed into the test wells. In the next step, 100 μL of compounds ranging from 1.9 to 125 g/mL were poured into the wells. While suspensions of microorganisms were used as a positive control in the study, TSB containing 1% glucose was used as a negative control. Plates were incubated for one day at 36.5 °C. After the plates were emptied, they were washed three times with 300 μL of deionized water. The wells were dried for half an hour. The next step was staining with 0.1% (*w/v*) violet dye, cleaning three times with deionized water, and drying at room temperature. The crystal violet adhering to the wall of the wells was dissolved with 95% ethanol, and the optical density was measured at 570 nm. Antibiofilm activities of compounds were measured at sub-MIC values using the equation below, where ODc and ODb stand for the compound's and the broth's respective absorbances [73]. The test was run three times, and the mean value was calculated. Biofilm preventing (%) = {(ODb − ODc)/ODb} × 100.

### *3.4. Theoretical Methods*
### 3.4.1. Molecular Docking

The target macromolecules were acquired in PDB format from https://www.rcsb.org/, accessed on 20 February 2023 (PDB ID: 1kzn and 5v5z) [74,75]. All the molecular docking performances were achieved by AutoDock 4.2 [76]. The macromolecules were cleaned from the water, and Kollman charges and the polar hydrogens were used during the performances. The ligand molecules with Gasteiger charges were randomly positioned for starting, and Lamarkian genetic algorithms were used [77].

### 3.4.2. DFT-Based Calculation Methods

The geometry optimizations were performed using ORCA package version 4.0 [78,79], and def2-SVP def2-SVP/J basis set and BP86 functional were used during the procedures [80]. tightscf and KDIIS SOSCF were used as the restriction of the calculation [81,82]. The global reactivity descriptors, IP (ionization potential), EA (electron affinity), χ (electronegativity), η (chemical hardness), S (global softness), and ω (electrophilicity index), were determined according to Koopmans Theorem [83]:

$$IP = -E_{HOMO} \tag{1}$$

$$EA = -E_{LUMO} \tag{2}$$

$$\chi = -\frac{I + A}{2} \tag{3}$$

$$\eta = \frac{I - A}{2} \tag{4}$$

$$S = \frac{1}{2\eta} \tag{5}$$

$$\omega = \frac{\mu^2}{2\eta} \tag{6}$$

## 4. Conclusions

We have shown that the reaction between five benzimidazolium salts substituted with methylaryl moieties and $Ag_2O$ in dichloromethane readily forms the corresponding silver(I) complexes, which were fully characterized using spectroscopic methods for one of them by a single-crystal X-ray diffraction study. These newly synthesized organometallic species were screened for in vitro antimicrobial and antibiofilm activities. The complexes provide antimicrobial activities, with MIC as low as 1.9 μg/mL, against *Staphylococcus aureus* and *Candida albicans* strains, which were similar in the range of standard drugs Ciprofloxacin and Fluconazole. The antimicrobial activities of the compounds were also found to be quite effective for other microorganisms. Finally, these silver complexes were shown to be very efficient in antibiofilm activity with up to 92.9% biofilm inhibition at 1.9 μg/mL on *Escherichia coli*. According to the molecular docking results, complex **9** has the best binding affinity with strong H-bonds against *Escherichia coli* DNA gyrase, while the highest binding constant was calculated for complex **10** against CYP51 from the pathogen *Candida albicans*. These results were in agreement with the in vitro experiments carried out on *Escherichia coli* and *Candida albicans*. Furthermore, the chemical reactivity descriptors were determined, and the results emphasized that LUMOs of complexes **6**, **7**, and **8** were located on the benzimidazole ring, while all HOMOs were located on the chloride atom. According to the descriptors and in agreement with in vitro tests, complex **10** was found to be the most reactive drug.

The present study has highlighted the importance of the *N*-heterocyclic carbene substituents on the antimicrobial activity of silver complexes, and the optimization of these drugs will be the subject of future studies.

**Supplementary Materials:** The following supporting information can be downloaded at https://www.mdpi.com/article/10.3390/inorganics11100385/s1, Characterizing data of silver complexes **6–10** ($^1$H and $^{13}$C NMR, FT-IR, and LC-MS spectra) and interaction details of Ciprofloxacin, Fluconazole and complexes **6–10** with DNA gyrase of *Escherichia coli* and CYP51 of *Candida albicans*.

**Author Contributions:** Conceptualization, İ.Ö., N.G., D.S. and N.Ş.; methodology, C.Ç., U.T., N.Ö., E.Ü. and N.Ş.; software, E.Ü. and N.Ö.; validation, C.Ç., U.T., N.Ö., N.Ş., D.S. and E.Ü.; formal analysis, N.Ş., E.Ü., C.Ç., U.T. and N.Ö.; investigation, N.Ş., C.Ç. and U.T.; writing—original draft preparation, D.S., N.Ş., E.Ü., C.Ç., U.T. and N.Ö.; writing—review and editing, D.S, N.Ş. and E.Ü.; supervision, İ.Ö, N.G, D.S, N.Ş. and E.Ü.; project administration, N.Ş.; funding acquisition, N.Ş. and N.Ö. All authors have read and agreed to the published version of the manuscript.

**Funding:** This research was funded by the Technological and Scientific Research Council of Turkey, project number TÜBİTAK-3001(118R045), and by Ondokuz Mayıs University, project number PYO.FEN.1906.19.001.

**Data Availability Statement:** The data presented in this study are available on request from the corresponding authors.

**Conflicts of Interest:** The authors declare no conflict of interest.

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
