# Peer review of "Benzimidazol-2-ylidene Silver Complexes: Synthesis, Characterization, Antimicrobial and Antibiofilm Activities, Molecular Docking and Theoretical Investigations"

_inorganics, doi:10.3390/inorganics11100385_

Round 1
Reviewer 1 Report
The manuscript presents the synthesis, X-ray diffraction and spectroscopic FT-IR, 13C and 1H NMR, and LC-MS characterization of five silver complexes (6-10) based on benzimidazole derivatives. DFT calculations were used to analyze the structure-reactivity properties by means reactivity descriptors from frontier molecular orbitals energies. The anti-microbial and biofilm formation inhibition were evaluated. Molecular docking calculations towards target proteins of Escherichia coli and Candida albicans were carried out. Complex 10 exhibited promising high antimicrobial activity comparable to fluconazole and ciprofloxacin drugs. The introduction of a silver atom in the complexes is tested to have a beneficial effect on antibacterial activity.
I consider original and relevant the topic included in the manuscript. The complexes based on benzimidazole derivatives including silver atom described in the manuscript are interesting compounds and they could have promising anti-bacterial properties. The computational techniques used for its structural and spectroscopic characterization are adequate and they provide relevant information about reactivity and anti-bacterial activity.
Therefore, my recommendation is Accept after Minor revisions. I suggest some changes and improvements as follows:
Minor revisions:
2. Results and Discussion
In the 2.2. Single-Crystal X-ray Diffraction Studies section, page 4, line 145, the distance “Ag-Cl” is repeated in the phrase, is it correct? I find this confuse.
In the 2.4. Molecular Docking section, page 7, line 215, “the recorded consensus scores in the range 5.55-2.41” mentioned, have any units? What does the term “consensus score” refer to? Is it comparable to a “binding energy” value?
Page 7, line 237, in phrase: “All molecules except 8 have higher binding affinity than Ciprofloxacin, but the H-bonds of the molecules are not as effective as Ciprofloxacin (see Supplementary Materials), and the phrase in Page 8, line 271, “All binding affinities were higher than calculated for Fluconazole (see Supplementary Materials).”; I find confuse these sentences because effectively the value of compound 8 has the higher binding affinity value (-6.21 kcal/mol) than ciprofloxacin (-6.89 kcal/mol), but in the case of fluconazole (-5.65 kcal/mol), all compounds 6-10 have smaller values than fluconazole (-9.26 to -7.26 kcal/mol). Authors must correct this contradiction, please.
Page 8, line 260, Figure 4 and Figure 5 should be improved, because they have very small texts and the amino acids names and the 2D interactions are undistinguishable.
3. Materials and Methods
In the 3.4.1 DFT-Based Calculation Methods, page 13, line 459, the references for the basis sets def2-SVP def2-SVP/J should be provided.
4. Conclusions
Page 14, lines 481-482, in the phrase: “According to the molecular docking results, complex 9 has the best binding affinity with strong H-bonds while the highest binding constant was calculated for complex 10”, Is there any relationship between the binding affinity with the binding constant? Please explain it.
I consider English language and style are fine/minor spell check required.
Author Response
We thank this referee for his/her comments.
* “In the 2.2. Single-Crystal X-ray Diffraction Studies section, page 4, line 145, the distance “Ag-Cl” is repeated in the phrase, is it correct? I find this confuse”.
They are not same. The first is Ag-Chlorine bond while the second is Ag-Carbon.
The text was modified.
* “In the 2.4. Molecular Docking section, page 7, line 215, “the recorded consensus scores in the range 5.55-2.41” mentioned, have any units? What does the term “consensus score” refer to? Is it comparable to a “binding energy” value?”
Patil et al (https://doi.org/10.1016/j.bioorg.2019.103217) use this term in their manuscript. They define the term as “Consensus Score integrates a number of popular scoring functions for ranking the affinity of ligands bound to the active site of a receptor and reports the output of total score”. This score includes many other scores such as crash-score, polar score, d-score, chem score etc. I understood that each score evaluates the particular interaction type of intermolecular interactions. We can generally define the binding energy as “the sum of all the intermolecular interactions between the ligand and the target”. We could not directly compare the quantitative results since Patil et al used different software and different interaction terms. However, the present manuscript is useful for emphasizing the importance of the target molecule and N-bonded molecules.
Unit kcal/mol was added
* “Page 7, line 237, in phrase: “All molecules except 8 have higher binding affinity than Ciprofloxacin, but the H-bonds of the molecules are not as effective as Ciprofloxacin (see Supplementary Materials), and the phrase in Page 8, line 271, “All binding affinities were higher than calculated for Fluconazole (see Supplementary Materials).”; I find confuse these sentences because effectively the value of compound 8 has the higher binding affinity value (-6.21 kcal/mol) than ciprofloxacin (-6.89 kcal/mol), but in the case of fluconazole (-5.65 kcal/mol), all compounds 6-10 have smaller values than fluconazole (-9.26 to -7.26 kcal/mol). Authors must correct this contradiction, please.”
Corrections were made
* “Page 8, line 260, Figure 4 and Figure 5 should be improved, because they have very small texts and the amino acids names and the 2D interactions are undistinguishable.”
The bigger size illustration of the 2D interactions were added in Figures 4 and 5 as well as in Supplementary Materials.
* “In the 3.4.1 DFT-Based Calculation Methods, page 13, line 459, the references for the basis sets def2-SVP def2-SVP/J should be provided.”
The reference is embedded in the text and Ref. list as Ref. 80.
* “Page 14, lines 481-482, in the phrase: “According to the molecular docking results, complex 9 has the best binding affinity with strong H-bonds while the highest binding constant was calculated for complex 10”, Is there any relationship between the binding affinity with the binding constant? Please explain it.”
Thanks for the great attention of the reviewer. The degree of binding of the ligand with the protein refers to the binding affinity. The energy released due to the bond formation, interaction of the ligand and protein is termed in binding energy. But we simply wrote a deficient sentence. The sentence was corrected as “According to the molecular docking results, complex 9 has the best binding affinity with strong H-bonds against Escherichia coli DNA gyrase while the highest binding constant was calculated for complex 10 against CYP51 from the pathogen Candida albicans.”
Reviewer 2 Report
The manuscript “Benzimidazol-2-ylidene Silver Complexes: Synthesis, Characterization, Antimicrobial and Antibiofilm Activities, Molecular Docking and Theoretical Investigations.” by Tutar et al. reports studies about five N-heterocyclic carbene silver(I) complexes. The complexes were characterized by FT-IR, 1H and 13C NMR spectroscopy and LC-MS spectroscopy. Their biological properties were tested against S. aureus, E. faecalis, E. coli, A. baumannii and C. albicans. Moreover Molecular docking suggested interactions with DNA gyrase of Escherichia coli and CYP51 from Candida albicans.
The novelty of this paper is not very high.
The authors have been engaged in this research activity for long time and have already published many papers on this topics. Some of the authors have been published very recently a similar paper (Tutar, U.; Celik, C.; Sahin, N. Allyl Functionalized Benzimidazolium-Derived Ag(I)-N-Heterocyclic Carbene Complexes: AntiBiofilm and Antimicrobial Properties. Pharm. Chem. J. 2022, 56, 54–60). The described compounds differ by only a methyl group on the NHC ligand. I believe the authors should compare the antimicrobial and antibiofilm activity with the previously reported compounds. Unfortunately, the comparison is not immediate because in this paper the authors use different units of measurement (mmol/mL vs mmg/mL). I strongly suggest comparing the compounds presented with those previously reported in order to obtain more information on the SAR of NHC ligands.
In a previous paper (Üstün, E. et al. Synthesis, characterization, antimicrobial and antibiofilm activity, and molecular docking analysis of NHC precursors and their Ag-NHC complexes. Dalton Trans. 2021,50, 15400–15412) the same authors observed the interactions between the complexes with Quorum-Quenching N-Acyl Homoserine Lactone Lactonase and the E. coli and B. anthracis Malate Synthase A. Why the authors changed their target? I believe in the paper they should discuss this point.
Furthermore, the conclusion could be improved by underlining the novelty of the obtained results
Minor point:
The introduction must be improved citing some reviews about NHC silver complexes useful for readers to better place this work in the current literature. Three examples are: Kascatan-Nebioglu, A.; Panzner, M.J.; Tessier, C.A.; Cannon, C.L.; Youngs, W.J. N-Heterocyclic carbene–silver complexes: A new class of antibiotics. Coord. Chem. Rev. 2007, 251, 884–895. Patil, S.A. et al. N-heterocyclic carbene-metal complexes as bio-organometallic antimicrobial and anticancer drugs, an update (2015–2020). Future Med. Chem. 2020, 12(24), 2239–2275. Ronga, L.; Varcamonti, M.; Tesauro, D. “Structure–Activity Relationships in NHC-Silver complexes as antimicrobial agents”. Molecules 2023, 28, 4435.
Author Response
We thank the referee for his/her insightful remark and careful reading of the manuscript.
* “In a previous paper (Üstün, E. et al. Synthesis, characterization, antimicrobial and antibiofilm activity, and molecular docking analysis of NHC precursors and their Ag-NHC complexes. Dalton Trans. 2021,50, 15400–15412) the same authors observed the interactions between the complexes with Quorum-Quenching N-Acyl Homoserine Lactone Lactonase and the E. coli and B. anthracis Malate Synthase A. Why the authors changed their target? I believe in the paper they should discuss this point.”
In the previous paper you pointed out, we studied Ag-NHC type complexes as the draft you reviewed, we want to analyze the interactions with different type of target molecules. We cited this paper as Ref. 12 and we also added a new sentence that emphasize the meaning of the paper.
* “Furthermore, the conclusion could be improved by underlining the novelty of the obtained results.”
The conclusion was modified.
* “The introduction must be improved citing some reviews about NHC silver complexes useful for readers to better place this work in the current literature. Three examples are: Kascatan-Nebioglu, A.; Panzner, M.J.; Tessier, C.A.; Cannon, C.L.; Youngs, W.J. N-Heterocyclic carbene–silver complexes: A new class of antibiotics. Coord. Chem. Rev. 2007, 251, 884–895. Patil, S.A. et al. N-heterocyclic carbene-metal complexes as bio-organometallic antimicrobial and anticancer drugs, an update (2015–2020). Future Med. Chem. 2020, 12(24), 2239–2275. Ronga, L.; Varcamonti, M.; Tesauro, D. “Structure–Activity Relationships in NHC-Silver complexes as antimicrobial agents”. Molecules 2023, 28, 4435.”
The article Coord. Chem. Rev. 2007, 251, 884–895 was already cited (Ref. 21).The two articles were added in the references (Ref. 14 and 15).